# Let Us Take It into Our Own Hands: Patient Experience during the COVID-19 Pandemic

**DOI:** 10.3390/ijerph192114150

**Published:** 2022-10-29

**Authors:** Martina Baránková, Katarína Greškovičová, Bronislava Strnádelová, Katarina Krizova, Júlia Halamová

**Affiliations:** Institute of Applied Psychology, Faculty of Social and Economic Sciences, Comenius University in Bratislava, 821 05 Bratislava, Slovakia

**Keywords:** consensual qualitative research, coping, COVID-19 pandemic, healthcare, patient experience

## Abstract

The COVID-19 pandemic introduced new health situations for patients and health professionals alike and, with them, opportunities to study these new patient experiences, gain insights into changed healthcare practices, and propose potential new healthcare solutions. The aim of our study was to explore how people coped with their health issues during the pandemic. We utilized a consensual qualitative analysis. The convenience sample that was gathered online through social media comprised 1683 participants with a mean age of 31.02 years (SD = 11.99). The 50 participants from the convenience sample who scored the highest on subscales of the COPE inventory were selected for in-depth interviews. In-depth interviews with 27 participants from the convenience sample who reported a health issue were analyzed. The final sample in our study therefore comprised 17 women (63%) and 10 men (37%) with a mean age of 28.35 years (SD = 9.31). The results showed that behavioral coping with health problems was mentioned across all participants’ accounts. However, participants facing a health issue during the COVID-19 pandemic mostly relied on their own self-help instead of on healthcare services. They utilized healthcare services only when absolutely necessary. Furthermore, the participants had two main sources of resilience: themselves and other people.

## 1. Introduction

The beginning of the COVID-19 pandemic brought turbulent changes in the healthcare system, both for patients and medical providers. Overcrowded hospitals and ambulances could not keep up with the influx of patients, and as a result, the work conditions in healthcare were severely impacted, and health workers’ mental health suffered. COVID-19 emerged as a unique and independent risk factor for the stress experienced by medical personnel [1]. Uncertainty and mistrust toward pandemic measures and later toward the COVID-19 vaccines and their efficacy also circulated among nonphysician healthcare workers and employees of the hospitals, especially those who were not infected by COVID-19 [2]. Mental health experts, such as psychiatrists and psychologists, also experienced high levels of personal and work-related stress during the pandemic [3]. The conditions of high stress, high demand, and low resources uncovered several limitations of healthcare systems in different countries. For example, in Slovakia, a clear need to reform the healthcare system was identified by uncovering the link between crisis leadership and team performance [4]. Therefore, the patient experience with health services was significantly different during COVID-19 in comparison to prepandemic times. Even though Slovak politicians implemented some of the harshest precautions in Europe during the COVID-19 pandemic [5], there were still periods of time for, example February 2021, when Slovakia had some of the worst death ratios and numbers of hospitalized patients with COVID-19 per capita in the world [6]. As a consequence, a mass resignation [7] of approximately 2000 doctors at the beginning of the October 2022 also reflected the state of the Slovak healthcare system and its ability to deal with the situation.

### 1.1. Patient Experience

The patient experience is defined via narrative synthesis as “… more than satisfaction, continuum of care, focus on expectations, individualized care and tailored services, and that patient experience is commonly aligned patient-centred care principles,” [8] (p. 12). The same team of researchers enriched this definition later by specifying the patient experience to be the sum of the interactions between care team members, family members, care partners, and context [9]. The patient experience is shaped by an organization´s culture and directly impacts the experiences of both the patients and the workers. Patient perceptions are influenced by the patient experience in the way healthcare organizations are seen and talked about. The patient experience is beyond the distinct boundaries of healthcare and across the continuum of care [9].

It is important to differentiate between *patient experience and patient satisfaction*. Both terms are used interchangeably, and there is a lack of agreement about their meanings [10]; however, these two terms likely should not be considered interchangeable [11]. While patient experience describes what happened from the patient´s perspective, patient satisfaction likely reflects patient’s expectations rather than the quality of healthcare. Patient satisfaction can, therefore, be classified as being one of the manifestations of patient experience [12].

This is supported by the data collected from 21 European countries that showed patient experience to be significantly associated with patient satisfaction and to explain approximately 10% of the variation in the patient satisfaction construct [13].

In his narrative review of the literature, Zakkar [12] synthesized the patient experience into two main domains: the determinants of patient experience and the manifestations of patient experience. Five determinants of patient experience were identified: the experience of illness, the patient’s subjective influences, the quality of healthcare services, the health system’s responsiveness, and the politics of healthcare. There were also two manifestations: patient satisfaction and patient engagement. The patient experience is thus a complex multidimensional phenomenon that encompasses different viewpoints accounting for differing stakeholders’ expectations and lived experiences with healthcare [12].

The patient experience is positively associated with both clinical effectiveness and patient safety, and therefore it is one of the pillars of quality healthcare [14,15]. The subjective experiences of patients and patient observations and their valid and reliable measurement have a strong potential to inform the public and healthcare stakeholders about potential effective healthcare improvements [16,17,18,19,20].

Several possible strategies can be found in the literature that address how to improve the patient experience, for example, by increasing the involvement of nurses in the healthcare process, given that nurses spend a lot of time with patients [21]. The engagement of nurses and the optimization of their communication with patients could lead to a direct positive impact on the patient experience [21]. Additionally, better communication with patients was also found to have an impact on the work satisfaction of healthcare professionals [22]. Of note are recent technological improvements in healthcare that create a high demand on the healthcare providers’ time due to their technological complexity and result in even less opportunities to focus on effective and empathetic patient communication [12].

### 1.2. Overview of Patient Experience Research during COVID-19

As mentioned, the patient experience is associated with patient satisfaction as well as with clinical effectiveness and patient safety. The results from a qualitative review of patient experience with emergency department services revealed that the main patient experience themes were organized around the needs of patients. These core themes included communication, emotions, competent care, physical and environmental needs, and waiting needs [15]. Specifically, for COVID-19 patients, five main categories or priorities emerged from the patients’ perspective: access to desirable care and comfort services; access to education and information from credible sources; access to specialized care; support social needs; and the need for deep emotional interactions [23].

Studies of the patient experience with participants who suffered from long COVID-19 described participants having unpredictable physical and psychological symptoms and being dissatisfied with their interactions with healthcare teams [24]. People with long-term respiratory conditions, such as asthma, were affected by social distancing measures and reported disruptions to care, difficulties accessing healthcare services, anxiety, loneliness, and worries about their own or their family members’ health [25].

A somewhat unique situation that arose during the COVID-19 pandemic was the long-term postponement of planned surgeries. Patients experiencing the postponement of their cardiac and vascular surgeries suffered emotional and psychological distress; their surgeries were a way to escape their symptoms; patients reported more fear of coronavirus than of dying of a heart attack [26].

The emergence of a novel patient experience due to COVID-19 led to new strategies to meet patient needs. For example, telemedicine has become a commonly used tool for medical providers [27,28]. Distant methods of healthcare delivery, such as phone calls or video calls, were considered necessary in order to maintain safety by reducing in-person visits during the pandemic while also providing effective patient care. Telehealth is appropriate to use for patients with both general health issues and specific health conditions [27,29,30]. Moreover, both patients and clinicians seem to appreciate the telecontact [29]. Patients appreciate telemedicine’s similarities to face-to-face encounters, since they prefer to see the clinician’s face, but they also consider it more convenient than an in-person visit. Similarly, patients also reported a preference to continue with regular check-ups via telemedicine [26]. Furthermore, clinicians seem to be open to its use and are satisfied with the continuous care of their patients in a virtual/online mode.

Distant delivery of medical care was also proposed as a necessary component in the care of psychiatric patients, whose symptoms often worsened during the pandemic [31]. The suggestions for care improvement for psychiatric patients included shifting care to telemedicine and safety-focused community-based psychiatry that would, for example, tighten admission criteria, shorten the length of hospital stays, suspend some group activities, and limit visitors [31]. It is likely that, in comparison to the symptoms of patients with physical health conditions, the symptoms of patients with psychological disorders were more impacted by the pandemic, and therefore psychological and psychiatric patients are of in dire need of additional psychological/psychotherapeutic support [32]. Boissy [33] pointed out that person-centered care continues to be important in post-COVID-19 times, although healthcare delivery methods might have changed.

Given the COVID-19 crisis, the call for better healthcare practices that are informed by the patient experience is even more urgent, and more research data are needed to make informed suggestions about improvements in healthcare.

## 2. The Aim of the Current Study

The COVID-19 pandemic introduced many novel health situations for both patients and health professionals. These novel situations give us an opportunity to study new patient experiences, gain insights into changed healthcare practices, and propose potential new healthcare solutions.

The aim of our study is, therefore, to explore how people cope with health issues during a pandemic by utilizing the patient experience perspective.

## 3. Methods

### 3.1. Qualitative Research Team

Five members of the core research team who are assistant professors in psychology coded the data, and one full professor of psychology audited the coding. All members of the team had previous experience with consensual qualitative research.

### 3.2. Research Sample

The convenience sample, gathered online through social media, comprised 1683 participants (67% women, 32.35% men, and 0.65% did not report their gender) with a mean age of 31.02 years (SD = 11.99). The participants’ ages ranged from 18 to 77 years. The respondents were of Slovak nationality, signed an informed consent, and completed an online questionnaire with sociodemographic questions and the COPE inventory items [34]. From the convenience sample, we selected fifty participants with the highest scores on subscales of the COPE inventory [35] because we were interested in the most adaptive coping strategies during the pandemic. A cut-off score of 10 points served as a benchmark for highly adaptive coping. High-scoring participants were invited for in-depth interviews and received a EUR 50 voucher as an incentive. There were 21 men (42%) and 29 women (58%) with a mean age 28.71 years (SD = 9.42) who met the cut-off score. Out of these 50 in-depth interviews, 27 interviews included a report of a health issue. These reports were from 17 women (63%) and 10 men (37%) with a mean age of 28.35 years (SD = 9.31). For the education level, one participant (3.7%) finished primary education, ten participants finished (37%) secondary education, three participants (11.1%) received bachelor´s degrees, and nine (33.3%) received master´s degrees. There were thirteen students (48.2%), one participant was on parental leave (3.7%), six were employed (22.2%), and two were unemployed (7.4%). There were also some missing data: five participants (two men and three women) did not report their age, education level, or employment status (for more detail see Table 1). Data were collected in accordance with the ethical standards of the institutional and/or national research committee and in accordance with the 1964 Helsinki Declaration and its later amendments or comparable ethical standards. The study´s protocol was approved by the Ethical Committee of the Faculty of Social and Economic Sciences at Comenius University in Bratislava.

### 3.3. In-Depth Interviews

#### Data Analysis: Consensual Qualitative Research

Before the analysis, all researchers reflected on their expectations about the participants’ coping during the pandemic in writing to make them aware of their potential bias. Then, the transcribed interviews were analyzed using the consensual qualitative research method (CQR) [36]. This method was chosen because it allowed for the exploration of inner experiences related to complex phenomena, and thus it was well-suited to help us understand complex patient experiences during the pandemic. In the CQR approach [36], participants are considered the experts who inform the researchers about their experiences. An essential part of the CQR method is arriving at a consensus about data interpretation between the core team members and between the team and the auditor. Reaching a consensus also allows for the minimization of the researchers’ biases.

In accordance with the CQR method (adapted from [36]), our research team first identified the relevant parts of the participants’ narratives, which were related to a health issue during the pandemic. Subsequently, the core team members created domains independently of each other and compared them during a discussion, which resulted in a consensual solution. The domains were then checked across multiple cases. The research team then constructed subdomains, categories, and subcategories within individual cases and conducted a cross-case analysis. The auditor reviewed the analyzed the domains, subdomains, categories, and subcategories of cross-cases; suggested revisions; and consulted with the team. The analysis was completed for all cases at once. Last, we created a typical story that reflected dealing with a health issue during the pandemic. According to Hill et al. [36], general categories are those reported by all participants, and typical categories are those reported by more than the half of the participants. The categories that were included only in one case were considered unique categories. In the final step of our analysis, we received feedback from the research participants about the results.

## 4. Results

The CQR analysis [36] of 27 interviews that included information about the participants’ health issues yielded 644 quotations coded with 208 unique codes. In total, 2 main domains, 4 subdomains, 16 categories, 28 subcategories, and 13 characteristics emerged from the data. The most frequent domain was dealing with health issues on one’s own, with 576 quotations (89.5%), followed by the dealing with health issues by utilizing healthcare providers domain, with 68 quotations (10.5%).

The quotations were divided into two main domains, the first with an emphasis on dealing with health issues on one’s own and the second with a focus on dealing with health issues by utilizing healthcare providers. The COVID-19 pandemic introduced novel health situations and novel strategies for how to cope with them. In our data, coping strategies focused on one´s own efforts were represented more than strategies focused on obtaining medical assistance.

### 4.1. Domain of Dealing with Health Issues on One’s Own

The domain of dealing with health issues on one’s own refers to a specific form of coping with health issues that the participants were able to recognize and address themselves in multiple ways. Three subdomains emerged in this domain: emotion-focused coping with health issues, cognitive coping with health issues, and behavioral coping with health issues (see Table 2 for the whole categorization and participant citations).

Emotion-focused coping with health issues was further divided into two categories: emotional experiencing and emotional processing. The participants reported experiencing and processing a lot of emotions that emerged for them while they were dealing with a health issue. The category of emotional processing did not yield further information; however, the emotional experiencing category included two subcategories: positive emotional experience and negative emotional experience. Among the negative emotions the participants experienced were feelings pain, sadness, regret, anger, boredom, hopelessness, fear, and despair. However, the participants also reported having positive emotions that were characterized by feelings of hope and joy and by savoring.

The second subdomain, cognitive coping with health issues, consisted of four categories: reflecting on the health issue, description of symptoms, acceptance of the health issue, and attention regulation. The category acceptance of the health issue represented the process of acceptance that the participants went through in order to accept the reality of their health situation. The category reflecting on the health issue included three subcategories: the realization of having a health issue, the processing of one’s health issue, and overwhelmed by health issues. The participants that realized they were dealing with a health issue described specific ways in which they considered their health limitations.

The category description of symptoms also included three subcategories: somatic symptoms, psychological symptoms, and psychiatric symptoms. Participants were able to recognize and describe their symptoms in detail. Another category in this subdomain, attention regulation, referred to the participants’ attempts to deal with their health reality by either distorting the facts of their health condition, avoiding health-related triggers, or consciously directing their attention elsewhere and paying attention to more pleasant stimuli.

The last subdomain, behavioral coping with health issues, consisted of six categories: information seeking and sharing, ways of coping, supporting others during their health issues, receiving support from others during one’s health issue, mutual support during health issues, and supporting oneself during health issues. The category information seeking and sharing referred to the participants’ efforts to actively search for and share information about the illness with their loved ones in order to cope with the pandemic. In the ways of coping category, the participants described different behavioral strategies that they used to deal with their health condition, for example, by limiting contact with others, spiritual coping, adhering to regulations, actively learning to cope, changing their habits, and looking for solutions. The participants regulated their contacts with others in two ways: closing themselves off to the outside world and being intentionally selective about who they spent their time with. Some participants experienced a decrease in interest in maintaining relationships, whereas others limited contact with their loved ones to protect them from the infection.

Finally, the category supporting others during their health issues described our participants’ efforts to help others with their health issues, for example, by offering practical help or by taking care of others. Similarly, others also offered help to our participants by providing practical help and caring support. In both cases, practical help meant that the help focused on day-to-day tasks and the caring support focused on activities that contributed to one’s well-being as they were dealing with a health issue. The participants also engaged in mutual support when they reciprocated the helping behaviors, which emerged as a separate category. Furthermore, our participants also described how they served as a source of support for themselves, for example, when they focused on taking care of their physical and mental health.

### 4.2. Domain of Dealing with Health Issues by Utilizing Healthcare Providers

The second domain, dealing with health issues by utilizing healthcare providers (Table 3), describes how our participants dealt with their health condition with the help of the healthcare system during the pandemic. Even though we separated the two main domains in our tables, these two domains were connected and were often intertwined, meaning that the participants used various strategies to address their health issues on their own and simultaneously sought help from a medical provider.

The subdomain of healthcare solutions, which emerged from the data, included four categories: the use of healthcare services, functional healthcare services, dysfunctional healthcare services, and attitudes toward healthcare services. The participants who utilized healthcare services did so by choosing a medical provider, going to see them, and contacting them, when necessary, via a phone call. The participants verbalized that sometimes the healthcare system worked well and they received care that they were happy with. However, sometimes they were frustrated with the lack of services or with the low quality of the provided services. Finally, the participants expressed their attitudes toward healthcare services that were characterized by either trust and confidence in their medical providers or doubts about the provided care, which were associated with feeling unsafe and fearful.

### 4.3. General Patient Experiences during the COVID-19 Pandemic

According to the CQR [36], general experiences are those reported by all participants. In our data, only behavioral coping with health problems was mentioned across all participants’ accounts. However, there were several codes, characteristics, subcategories, and categories that were mentioned by more than half of our sample (more than 14 participants), and thus they would be considered typical patient experiences.

### 4.4. Typical patient experiences during the COVID-19 Pandemic

During the pandemic, our participants/patients reported dealing with several health issues, which they addressed by either trying to help themselves, turning to medical professionals for help, or both. All participants used behavioral coping strategies, and more than half used strategies focused on cognitive and emotional coping. In terms of frequency, our participants were the least likely to seek professional help as a primary way of coping with their health issue.

Our patients experienced negative emotions, with feelings of fear being the most common. Our participants then utilized cognitive coping and reflected on their health issues in order to search for solutions. That meant that the participants recognized their health situations and tried to process them on a conscious level. They also tried to consciously shift their focus to more pleasant stimuli, which meant that they spent energy on directing their cognitive effort away from the health-related situation. Furthermore, the participants had two main sources of resilience: themselves and other people. The participants described ways in which they relied on themselves to take care of their bodies and minds. However, our patients also experienced lots of help from their loved ones.

## 5. Discussion

Our goal for this study was to analyze patient experiences with health issues during the COVID-19 pandemic and identify ways of coping. An understanding of the patient experience provides a basis for the implementation of new strategies for healthcare by directly addressing the patient needs resulting from the patient experience during the pandemic.

Two main domains of patient experience with health issues emerged in this study: dealing with health issues on one’s own and dealing with health issues by utilizing healthcare. These domains are more interconnected than separated. However, the domain focused on participants dealing with their own health issues was dominant in our patients’ narratives, and therefore the utilization of healthcare was more supplementary and seemed to occur in unavoidable situations when lay people had no competencies to provide healthcare to themselves or their loved ones. The general patient experience consisted mainly of behavioral coping with health issues, which shows the human tendency to act, do something about a situation, and change it. Regarding cognitive coping, the participants typically tried to find solutions to deal with their health issues, and emotionally they experienced and had to process a lot of primarily negative emotions, especially fear. Their experience of a health issue was mainly accompanied by distinct negative emotions (e.g., fear, anger, pain, hopelessness, etc.), although the participants also described experiencing positive emotions such as joy or hope.

Receiving support from loved ones or engaging in reciprocal support with loved ones were also discussed by the participants as coping strategies. From the attachment theory perspective, in times of need, stress, or adversity, people turn to attachment figures for reassurance, support, and assistance. Moreover, according to the source of strength (SOS) model [37], through the function of a safe haven, attachment figures also help people with becoming stronger in the process of dealing with adversity.

Similar coping strategies were described in previous studies [15,23]: the participants’ focused on their emotional experience, their cognitive processing, and their need to communicate with others.

Our study showed that the participants preferred to take their health into their own hands when they predominantly engaged in the self-management of their health. Thus, they were active agents in their own healthcare. Patient engagement is an aspect of the patient experience that reflects the patients’ commitment to their health [12]. We believe that more research is needed in this area. There is evidence that patient engagement is critical in improving healthcare. However, research has so far seemed to focus on direct patient care strategies rather than on patient engagement strategies [38]. Patient engagement comes with patient empowerment when it comes to the knowledge and ability to make one’s own health-related choices [12]. One thing to consider here, however, is the boundary between self-help and professional help, which is not always clear in patient engagement. In order to clarify the boundary, more research related to patients’ experiences in various different health situations is needed.

The preference of our participants to utilize self-help strategies over healthcare services could be attributable to the nature of their health issues, their empowerment, and their engagement in their health issues or to the healthcare crisis created by the COVID-19 pandemic. The participants demonstrated more independence in their health self-care, more responsibility, and more awareness when it came to their health. Since the interviews we conducted with the participants were not specifically focused on their utilization of healthcare services, but rather on their overall handling of their health issue during the pandemic, it is possible that participants were more drawn to share their self-help strategies compared to the medical services they received. Regardless, their reports represent the continuum of care in health-related issues and the complexity of the patient experience. As Wolf et al. [9] mentioned, the patient experience is across the continuum of healthcare.

Another reason for taking things into their own hands could be attributed to the collapse of the healthcare system in Slovakia during the COVID-19 pandemic, as Slovakia had some of the worst death ratios and numbers of hospitalized patients with COVID-19 per capita in the world [6], along with the massive resignation [7] of doctors and nurses that were not able to bear the situation physically or emotionally.

Multiple suggestions for healthcare improvement could be made based on this study. First, our participants, similar to Moayed et al. [23], communicated the need for trustworthy information; therefore, it is important that professional medical information is readily available to the public. One issue, though, which was also confirmed by our participants’ accounts, is the doubt directed at medical professionals. Some antiscience and conspiracy information was reported by our participants and was described in the category attitude toward healthcare services. This result was corroborated by other research that highlighted the existence of conspiracy theories during and after the pandemic [39].

Second, the remote delivery of care, whether through telemedicine or phone calls, could bring more opportunities for people struggling with health issues, especially for those who are typically overlooked by the healthcare system, such as people with chronical conditions, mental health issues, aging needs, or special needs [12]. An offsite model of healthcare services could be beneficial for patients [29,31,40] as well as healthcare professionals. The usage of the offsite model of healthcare relies on communication between the patients and the healthcare providers, which can lead to improvements in patient–provider communication and subsequently to a more patient-centered approach [22,33]. However, it is important to keep in mind that the improvement in patient–provider communication also requires systematic changes in healthcare.

Third, a higher level of self-reliance when dealing with health issues and being a support system to oneself, in addition to receiving support from others, indicates that people are more likely to turn to their own resources when other help is either unavailable or perceived as unavailable. Previously, the patient experience was mainly explored in the context of ameliorating the soft skills of healthcare professionals [21,22]. We propose that the healthcare system could capitalize on patient resourcefulness and provide patients with effective self-help strategies that they could utilize at home when the healthcare system is overloaded. It could, for example, work to improve patients’ mental health by providing patients with distance delivery interventions that would be designed to expand their adaptive coping repertoire. We believe that improving patients’ mental health could have a positive effect on physical health, whether directly through lowering stress or indirectly through an increase in compliance with medical care.

The proposed solutions, however, depend on the severity of the health issue, patient symptom awareness, patient self-reflection and self-care, the accessibility of medical treatment, and the responsiveness to medical treatment. It is important to note that health issues are rarely isolated. They are embedded in intrapersonal, social, and health settings. Since it is the right of patients to be treated respectfully and empathetically regarding their health or nonhealth needs [41], considering all aspects, determinants, processes, and manifestations of the patient experience is crucial.

### Limitations

We are aware of several limitations in our study. First, during the in-depth interviews, our participants were asked to talk about their coping with the pandemic, which could have led them to focus on their self-care strategies more and on their healthcare system utilization less. Second, only residents of one European country (Slovakia) were interviewed, giving us patient experience reports that are limited to the healthcare pandemic measures of one country. Third, our study focused on those who managed the pandemic well from a coping perspective. Potentially disadvantaged groups such as elderly people, migrant workers, or chronically ill patients were not incorporated into the sample, which also brings the opportunity for further research to focus primarily on those groups. Elderly people may not have had sufficient access to healthcare when travel restrictions were implemented because they were not familiar enough with online healthcare services [42] (Liu et al., 2022). In China, some migrant workers must rely on self-help because they face severe ethnic discrimination [43] (Liu et al., 2022). They were afraid of going to the hospital and concealed their COVID-19-like syndromes to avoid forced evictions. This did not just happen among Chinese rural–urban migrants. The concealment of COVID-19 infection has been found in many other contexts [44] (O’Connor and Evans, 2022). For a broader view, disadvantaged groups also need to be incorporated into future research.

Finally, given that this study was conducted during the pandemic, the interviews were conducted via videoconferencing, which may have been perceived as a barrier and may have impacted the participants’ rapport with researchers [45].

## 6. Conclusions

The study explored how people coped with health situations during the pandemic by utilizing the patient experience perspective. The team of four researchers and one auditor analyzed the data using the consensual qualitative research method. A convenience sample of the fifty participants with the highest scores on subscales of the COPE inventory [35] were interviewed. The patients’ experiences were reflected by two main domains: dealing with health issues on one’s own and dealing with health issues by utilizing healthcare. The participants had to utilize novel healthcare strategies that adapted to the demands of the COVID-19 pandemic. For example, they were more independent in addressing their health issues, and they were more responsible and aware of the need to engage in self-care strategies. It is promising that our participants demonstrated resilience by being active agents in their own treatment. Patient engagement and patient empowerment might be, therefore, helpful in improving the patient experience. Our study contributed to the understanding of coping with health situations during the pandemic from the perspective of the patient experience. The results can contribute to the development of new policies in the Slovak healthcare system based on greater attention to self-care.

## Figures and Tables

**Table 1 ijerph-19-14150-t001:** Description of sample.

Age	Gender	Education Level	Employment Status
18	man	primary education	Unemployed
19	woman	secondary education	Student
20	man	secondary education	Student
20	woman	secondary education	Student
20	woman	secondary education	Student
21	man	secondary education	Student
21	woman	secondary education	Student
22	man	bachelor’s degree	Student
23	woman	secondary education	Student
23	man	secondary education	Student
23	man	bachelor’s degree	Student
25	woman	secondary education	Student
26	woman	bachelor’s degree	parental leave
27	woman	secondary education	Student
27	woman	master´s degree	employed or self-employed
28	woman	master´s degree	Student
36	woman	master´s degree	Unemployed
38	woman	master´s degree	employed or self-employed
38	woman	master´s degree	employed or self-employed
38	man	master´s degree	employed or self-employed
39	woman	master´s degree	employed or self-employed
46	woman	master´s degree	missing data
54	man	master´s degree	employed or self-employed

*Note* part from the participants displayed in the table, there were five participants that did not report their age, education level, or employment status.

**Table 2 ijerph-19-14150-t002:** Categorization of the domain of dealing with health issues on one’s own.

Subdomains	Categories	Subcategories	Characteristics	Examples
Emotion-Focused Coping with Health Issues	Emotional Experiencing	Positive Emotional Experience	Feelings of Hope	“When I was able to get the medicine, I was holding it in my hand, I felt so much hope that we finally got it.”
			Feelings of Joy	“He was refusing to go to the hospital. I was not sure whether he was going to make it, but he did. And without going to the hospital. I was so happy, it felt like winning the lottery.”
			Savoring	“Despite my broken leg, I was still able to enjoy the summer. One friend even took me to a beach party one night.”
		Negative Emotional Experience	Feelings of Pain	“I could not deal with the pain. It was such an emotional turmoil.”
			Feelings of Sadness	“I felt it so strongly, I was so sad about what happened.”
			Feelings of Regret	“I felt so much regret about not being able to say goodbye to her.”
			Feelings of Anger	“So I was fighting the anger a bit, not just a bit, I was fighting the anger.”
			Feelings of Boredom	“I was at home on a medical leave, and I was bored.”
			Feelings of Hopelessness	“I spent the whole month trying to get the medicine. And I felt so hopeless, because there were contradicting opinions whether the medicine can help.”
			Feelings of Fear	“I felt a knot in my chest, huge heaviness, as if someone was putting pressure on me. That’s how anxious I felt.“
			Feelings of Despair	“I felt so much despair. I have a lot of immunity-related health issues, such as asthma and different allergies, I was worried whether I would make it.”
	Emotional Processing			“I did not get used it, but I told myself, that even though it was not pleasant, I need to tolerate it.”
Cognitive Coping with Health Issues	Reflecting on the Health Issue	The Realization of Having a Health Issue		“I was depressed before, so I was able to recognize the symptoms during early onset. I was not following my sleeping routine, because I did not have to.”
		The Processing of One’s Health Issue		“I am really focused on thinking about it, I am processing. I am considering vaccination, but my father started having issues after being vaccinated, so that’s something to think about.”
		Overwhelmed by Health Issues		“I felt like my whole world was falling apart…”
	Description of Symptoms	Somatic Symptoms		“My health worsened, I started putting on weight, my eyesight got worse.”
		Psychological Symptoms		“I found it really hard to focus on school and on studying. I was not able to study.”
		Psychiatric Symptoms		“My (social phobia) condition worsened because I was not getting any practice talking to other people.”
	Acceptance of the Health Issue			“At last, we realized that it was what it was, and we needed to accept it.”
	Attention Regulation	Reality Bending		“On the other hand, I was trying to consciously ignore it. Five, six thousand people a day were getting sick during that time.”
		The Avoidance of Negative Triggers		“I made a conscious decision not to think about the possibility of my mother or other family member getting sick or not to think about how long the lockdown would last.”
		Shifting Attention Elsewhere		“Maybe my work with children, preparing the materials to teach them online helped me to redirect.”
Behavior-Focused Coping with Health Issues	Information Seeking and Sharing			“I am trying to educate others, explain the issue to them in hopes that having more information would help them to cope with things more rationally.”
	Ways of Coping	Limiting Contact with Others	Becoming Closed-Off	“I stopped reaching out to my closest friends. I felt very distant from everyone, my friends and family included.”
			Being Selective in Contact with Others	“I decreased my contact with people, especially the ones who had not have COVID. I did not want to expose them to it and I did not want to make others sick, especially my grandmother.”
		Spiritual Coping		“We are Christian, so our faith in God brought us some peace in this situation.”
		Adherence to Regulations and Responsibility		“Hand-washing, mask-wearing, complying with regulations. I did it all in an effort to be responsible.”
		Learning How to Cope		“strategies and coping skills, and I was learning to apply them.”
		Change in Habits		“I tried to do something new every day. That helped me to differentiate between days because otherwise I had trouble telling the days apart.”
		Looking for Solutions		“You just need to take matters into your own hands, just somehow deal with it.”
	Supporting Others During Their Health Issues	Giving Practical Help to Others		“We were able to get an oxygen machine for him to have at home because he refused to go to the hospital.”
		Taking Care of Others		“When she needed something, I bought it and brought it to her to the hospital.”
	Receiving Support From Others During One’s Health Issue	Getting Practical Help from Others		“I would find grocery bags left on the porch that were purchased and brought to us by our neighbours, grandparents, cousins, colleagues. They all took care of us.”
		Others Providing Care		“My work colleagues sent me flowers once, or sometimes someone sent a nice text message, and it really warmed my heart.”
	Mutual Support During Health Issues			“Our communication has intensified, as if there is now more space and love and some kind of mutual coexistence together…”
	Supporting Oneself During Health Issues	Taking Care of One’s Body		“Outdoor activities, primarily, such as hiking, biking, and others.”
		Taking Care of One’s Mind		“I was able to get myself together. I encourage myself to be more engaged in important areas of my life, such as at work or in relationships with my friends.”

**Table 3 ijerph-19-14150-t003:** Categorization of the domain of dealing with health issues by utilizing healthcare providers.

Subdomains	Categories	Subcategories	Examples
Healthcare Solutions	The Use of Healthcare Services	Choosing a Medical Provider	“I wanted to seek out a psychologist or psychotherapist due to my worsened mental health. Even issues from the past emerged that I thought were resolved.”
		Seeing a Medical Provider	“I started seeing my eye doctor because of this.”
		Calling a Medical Provider	“I was either able to find the information online or I called the practice and talked to the provider.”
	Functional Healthcare Services		“I had a fantastic primary care physician who would call me back more than once if I was not able to reach her earlier in the day.”
	Dysfunctional Healthcare Services		“It was very upsetting that I was taking such strong medication and I could not reach my provider. Finally, in April, I was able to get to them, but it took about three weeks.”
	Attitude toward Healthcare Services	Trust in Medical Providers	“I had trust in the doctor who performed my surgery.”
		Doubts about Healthcare Utilization	”I wanted to go home, I did not want to stay in the hospital. Actually, I was afraid to stay in the hospital, so I asked for early discharge and I left.”

## Data Availability

In order to comply with the ethics approvals of the study protocols, the data cannot be made accessible through a public repository. However, the data are available upon request for researchers who consent to adhering to the ethical regulations for confidential data.

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
