# Peer review of "Let Us Take It into Our Own Hands: Patient Experience during the COVID-19 Pandemic"

_ijerph, 2022, doi:10.3390/ijerph192114150_

Round 1

Reviewer 1 Report

Thanks for inviting me to review this manuscript. A very interesting, well-organised and insightful paper. I have only one major comment:

As the author indicated, people preferred self-help during the pandemic because they were “more independence in their health self-care, more responsibility, and more awareness when it came to their health”. But were there other possibilities that motivated them to do so? For example, older people may not have sufficient access to healthcare when travel restrictions were implemented because they were not familiar enough with online healthcare services (Liu et al., 2022a). In China, some migrant workers have to rely on self-help because they faced severe ethnic discriminations (Liu et al., 2022b). They were afraid about going to the hospital, they concealed their Covid-like syndromes to avoid forced evictions. This did not just happen among Chinese rural-urban migrants, the concealment of Covid infection has been found in many other contexts (O’Connor & Evans, 2022). Have you found anything similar in your investigation? Did anyone of your interviewees admitted that they had concealed their Covid-like symptoms and why? Those who concealed their symptoms would definitely rely on self-coping strategies but does this mean that they preferred this way? Could you discuss this more based on your observations?

Were there any socially disadvantaged participants in your study? Were their experiences and preferences of health coping very different from their better-off counterparts? It is a common practice to put some basic info about your interviewees in a Table in the methodology section. I am aware that you may not have such information but please consider provide what you have at hand. I believe the discussion section can be strengthened by talking about the differences between different populations groups, with a special focus on vulnerable groups. This is because quite a few studies have indicated that different population groups have been influenced by and responded to the pandemic and its interventions in very different ways (e.g., Amadasun, 2020; Purtle, 2020). However, the current manuscript lacks a bit of discussion about this.

Reference

Amadasun, S. (2020). COVID-19 palaver: Ending rights violations of vulnerable groups in Africa. World Development, 134(1), 1-2.

Liu, Q., Liu, Z., Lin, S., & Zhao, P. (2022a). Perceived accessibility and mental health consequences of COVID-19 containment policies. Journal of Transport & Health, 101354.

Liu, Q., Liu, Z., Kang, T., Zhu, L., & Zhao, P. (2022b). Transport inequities through the lens of environmental racism: rural-urban migrants under Covid-19. Transport policy, 122, 26-38.

O’Connor, A. M., & Evans, A. D. (2022). Dishonesty during a pandemic: The concealment of COVID-19 information. Journal of Health Psychology, 27(1), 236-245.

Purtle, J. (2020). COVID-19 and mental health equity in the United States. Social psychiatry and psychiatric epidemiology, 55(8), 969-971.

Author Response

Dear reviewer,

thank you for your quick and useful work. We really appreciate your comments. Please see our responses in the text below.

We assume that participants relied on self-help due to the inability of Slovak healthcare system to deal with the situation. See more added into the introduction of the article and into the discussion. We did not come across lying and concealing the symptoms in our interviews.

There are certainly other options related to the motivation of choosing health self-care during the pandemics. In our study, we did not focus on specific populations such as older adults, migrant workers or disadvantaged people. We worked with people who were good at managing their COVID-19 experience, therefore we did not reflect the experiences of specific groups that would go beyond the purpose of our research. We have incorporated your comment as a point for the future research. We added more info about our final sample.

Reviewer 2 Report

My review comments are attached. 

Author Response

Dear reviewer,

thank you for your quick and useful work. We really appreciate your comments. Please see our responses in the text below.

  • We specified the selection of participants for in-depth interviews
  • The outcome and contribution of the study are incorporated in the discussion and conclusion
  • Introduction already defines research gap, research aim and research question
  • COPE Inventory is specified
  • Qualitative study saturation is added
  • Our findings were discussed with other authors in discussion
  • Conclusion sumarizes the contribution of the study
  • Limitations are specified
  • Current findings from latest research were used in the study

Reviewer 3 Report

Thank you for an opportunity to comment on a mss entitled Let´s take it into our own hands: Patient experience during the COVID-19 pandemic. The mss is well written and engaging, and presents valuable information. A few queries for the authors:

1. Introduction - Can you provide information on the Covid-19 pandemic situation specifically in Slovakia? This will provide a good background to the study rationale and discussion of results.

2. Methods/Research Sample - Can you provide more information on the recruitment procedures and more details on the eligibility criteria for participants?

3. Table 1 - This table is difficult to read. Can the authors present the data in a landscape format?

4. Discussion - Can the authors provide more information how the study results can inform improvement of patents' experience during a pandemic (or longer-term following the Covid-19 pandemic) in Slovakia?    

Author Response

Dear reviewer,

thank you for your quick and useful work. We really appreciate your comments. Please see our responses in the text below.

  1. Thank you, more information about COVID-19 pandemics in Slovakia is added.
  2. More details about recruitment procedure and eligibility criteria are added.
  3. Table 1 is updated to a landscape format.
  4. More information about patients experience improvement in Slovakia is added.

Round 2

Reviewer 1 Report

Thanks for adding the sociodemographic information of your interviewees. But the authors failed to deal with my first comment in the previous report. 

In fact, your newly added table (table 1) makes it even more important to have a discussion about other populations especially those disadvantaged. Because your paper is about the patient experience, not the patient experience of better-off populations (higher educated people and students). Your sample is therefore seriously biased not because other patients may not experience the pandemic in a similar way. 

This is one of your major methodological flaws and adding a discussion about that is perhaps the most doable way to mitigate this flaw at this stage. 

You could revise and use my first comment in the previous report directly as part of the discussion but empirical findings based on the same context is strongly encouraged. 

It is the discussion section, what you need to do is to jump out of your study a little bit and think it through from a more general perspective.

Author Response

Dear reviewer,

thank you for your response. Mentioned issues were incorporated into the discussion part of our manuscript.